

# The fall of the mycobacterial cell wall: interrogating peptidoglycan synthesis for novel anti-TB agents

Cheng-Yu Chiang and Nicholas P. West

School of Chemistry and Molecular Biosciences, University of Queensland, Brisbane, Queensland, Australia

## ABSTRACT

Tuberculosis (TB) caused by *Mycobacterium tuberculosis* has been a threat to human health for thousands of years and still leads to millions of deaths each year. TB is a disease that is refractory to treatment, partially due to its capacity for in-host persistence. The cell wall of mycobacteria, rich in mycolic acid, is broadly associated with bacterial persistence together with antimicrobial and immunological resistance. Enzymes for the biosynthesis of bacterial peptidoglycan, an essential component of the cell wall, have been addressed and considered as appealing drug targets in pathogens. Significant effort has been dedicated to finding inhibitors that hinder peptidoglycan biosynthesis, many with demonstrated enzymatic inhibition *in vitro* being published. One family of critical biosynthetic enzymes are the Mur enzymes, with many enzyme specific inhibitors having been reported. However, a lesser developed strategy which may have positive clinical implications is to take advantage of the common structural and catalytic characteristics among Mur enzymes and to allow simultaneous, multiple Mur inhibition, and avert the development of drug resistance. *M. tuberculosis* relies on these essential Mur enzymes, with the best-known subset being Mur ligases, but also utilizes unique functions of atypical transpeptidases resulting in peptidoglycan peptide cross-linking beneficial to the bacteria's capacity for chronic persistence in humans. A systematic review is now needed, with an emphasis on *M. tuberculosis*. The urgent development of novel anti-TB agents to counter rapidly developing drug resistance requires a revisit of the literature, past successes and failures, in an attempt to reveal liabilities in critical cellular functions and drive innovation.

# INTRODUCTION

Tuberculosis (TB), a highly contagious disease, is caused by the bacteria *Mycobacterium tuberculosis* and remains a global health concern today. According to the WHO, 10.6 million people fell ill with TB in 2022 of which 1.4 million people lost their lives (*World Health Organization, 2023*). Treatment regimens are protracted and costly, even for drug susceptible cases. The persistence of *M. tuberculosis*, resulting in latent TB infection, dictates a long treatment regimen, leading to unwanted side effects, reduced patient compliance and drug resistance. Front-line treatment includes a biphasic strategy with four drugs (isoniazid, rifampicin, ethambutol and pyrazinamide) administered for an

Corresponding author
Nicholas P. West, n.west@uq.edu.au

initial 2-month intensive phase, and a subsequent 4-month continuation phase comprising isoniazid, rifampicin. Drug resistance in TB is primarily due to spontaneous chromosomal mutations of *M. tuberculosis* and subsequent selection (*Migliori et al., 2012*). The treatment success rate of multidrug-resistant (MDR) TB and extensively drug-resistant (XDR) TB, was reported as just 60% and 57% respectively in 2022 for the American region and it is projected to be lower globally (*World Health Organization, 2023*). According to 2022 guidelines an improved 6-month treatment regime composed of bedaquiline, pretomanid, linezolid and moxifloxacin or a 9-month all-oral treatment regime is recommended for MDR-TB patients. However, the evidence of efficacy for these modern regimes is limited and longer treatment is often required (*World Health Organization, 2022*). In cases which prove to be refractory to treatment, surgical resectioning of diseased lung remains an option (*World Health Organization, 2022*; *World Health Organization, 2023*). The cost of MDR-TB treatment is impossible for most of the patients in developing countries, ranging from 1,500 USD to 8,000 USD, and up to 200,000 USD in developed countries (*Akalu et al., 2023*; *Katrak, Wang & Barry, 2023*). These intensify the challenge of managing TB and novel therapeutic agents for TB are urgently needed.

The persistence of *M. tuberculosis* is in part due to the highly specialised mycobacterial cell wall. The mycobacterial cell wall complex is primarily composed with three components; mycolic acid, arabinogalactan and peptidoglycan (mAGP) (*Jarlier & Nikaido, 1994*). Mycolic acid, a very long chain fatty acid (60–90 carbons), forms the outermost layer of mAGP and acts as the major permeation barrier (*Brennan & Nikaido, 1995*; *Jarlier & Nikaido, 1994*). Arabinogalactan contributes to structural rigidity, permeation selectivity, and provide the anchor sites for mycolic acids and peptidoglycan (*Alderwick et al., 2015*). Peptidoglycan, onto which the arabinogalactan layer is covalently bound in mycobacteria, is present in all bacteria and serves as the essential component in maintaining cellular morphology and resistance to osmotic pressure, while maintaining flexibility for cell expansion and division (*Cabeen & Jacobs-Wagner, 2005*). It is well-acknowledged that the mAGP complex largely attributes the intrinsic resistance of mycobacteria (*Jarlier & Nikaido, 1994*; *Mdluli et al., 1998*; *Nikaido & Jarlier, 1991*).

The importance of peptidoglycan, and its absence from mammalian cells, makes it an attractive target for antibiotic development. The backbone of peptidoglycan consists of alternating N-acetyglucosamine (GlcNAc) and N-acetylmuramic acid (MurNAc) linked *via* ß(1 → 4) glycosidic bond. The pentapeptide side chain comprises L-alanine (L-Ala), D-glutamic acid (D-Glu), meso-diaminopimelic acid (m-DAP), and dimeric D-alanine (D-Ala-D-Ala), anchored on the lactyl group of MurNAc is responsible for the formation of a crosslinking bridge with the adjacent glycan strand. Inter-peptide crosslinking predominantly occurs between the carboxyl group of D-Ala at position four of one peptide and mDAP in the third position on the adjacent pentapeptide stem (3 → 4 linkage), establishing the rigid and mesh-like structure of peptidoglycan (Fig. 1) (*Raghavendra, Patil & Mukherjee, 2018*). In mycobacteria, a unique peptide linkage pattern can exist during non-replicative persistence or under environmental stress where linkage occurs between diaminopimelic acid of adjacent chains (*Lavollay et al., 2008*; *Wayne & Sohaskey, 2001*).

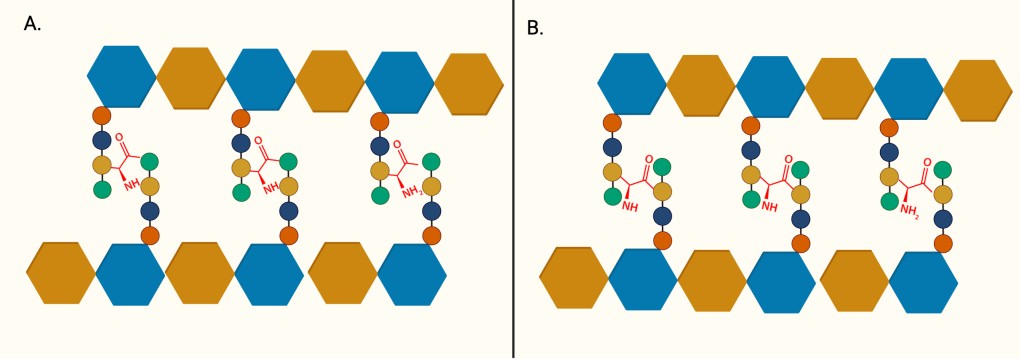

**Figure 1** **The structure of peptidoglycan crosslinking.** (A) Typical 4-3 interpeptide crosslinking bridge of bacterial peptidoglycan. (B) Atypical 3-3 interpeptide crosslinking bridge found in mycobacteria, particularly during persistent or non-replicating state. The blue hexagon represents UDP-MurNAc and the brown hexagon represents GlcNAc. L-Alanine (red circle), D-glutamic acid (yellow circle), D-alanine (green circle), and m-diaminopimelic acid (green circle) are represented in the peptide crosslinking bridge.

These unique features of mycobacterial peptidoglycan provide insights which might be exploited for clinical use in the future. This review summarises the published inhibitors for essential enzymes in peptidoglycan biosynthesis, including efforts to interrogate the atypical transpeptidases of *M. tuberculosis,* which can potentially act as the starting point of novel anti-tubercular therapeutic agents. We hope this review serves as a guide and a catalyst to prime beneficial strategies and future collaborative investigations between the broad, multi-disciplinary TB research community.

## SURVEY METHODOLOGY

The literature in scope with the topic of this review was assessed with PubMed, Google Scholar, and Web of Science using key terms: "Mur ligases", "Mur C ligase", "Mur D ligase", Mur E ligase", "Mur F ligase", "Mur X", "Mur G", "Mur J", "peptidoglycan biosynthesis", "bacterial cell wall biosynthesis" in combination with "inhibitors", "*Mycobacterium tuberculosis*, "drug development", "drug screening", "antitubercular", "antibacterial", "structure–activity relationships", "multitarget inhibitor", "antibiotics", "capuramycin", "ß-lactamase", "ß-lactam", "carbapenem" with "AND" and "OR" operators in search query to ensure comprehensive and unbiased literature search was conducted. The literature identified was evaluated by their pertinence to the topic and all related articles are thoroughly reviewed. The enzyme inhibition activity, antimicrobial activity and reference of compounds mentioned in the text are listed in Table 1.

### Synthesis of peptidoglycan

Peptidoglycan synthesis has been thoroughly studied in many organisms since its discovery in 1952 (*Park, 1952*). Nevertheless, the precise peptidoglycan synthesis pathway in *M. tuberculosis* remains ambiguous. The generally accepted assumption is that mycobacterial peptidoglycan is synthesised similarly to that of *Escherichia coli* due to the

**Table 1  List of identified compounds with the corresponding target, IC$_{50}$, MIC and reference.**

| ID | Target | IC$_{50}$ | MIC | Reference |
|----|--------|-----------|-----|-----------|
| 1 | MurC | *E. coli*: 42 nM | Not determined | *Reck et al. (2001)* |
| 2 | MurC | *E. coli*: 2.3 μM | Not determined | *Ehmann et al. (2004)* |
| 3 | MurC | Not specified 12–24 μM | *E. coli*: >65 μM *S. aureus*: 31–93 μM | *Sim et al. (2002)* |
| 4 | MurC | *E. coli*: 30.2 μM *K. pneumoniae*: 26.4 μM *P. mirabilis*: 41.4 μM | No antimicrobial activity | *Zawadzke et al. (2008)* |
| 5 | MurC | *E. coli*: 12 nM *P. aeruginosa*: <10 nM | *E. coli* ΔTO1C: 3.13 μM *P. aeruginosa-546*: 1.56 μM | *Hameed et al. (2014)* *Humnabadkar et al. (2014)* |
| 6 | MurD | *E. coli*: 0.7–9.0 μM | Not determined | *Horton et al. (2003)* |
| 7 | MurD | *E. coli*: 10 μM | No antimicrobial activity | *Tomašić et al. (2011)* |
| 8 | MurD | *E. coli*: 34 μM | *S. aureus*: 128 μg/mL *E. faecalis*: 128 μg/mL | *Zidar et al. (2011)* |
| 9 | MurD | *S. aureus*: 22.33 μM | *S. aureus*: 2 μg/mL *K. pneumoniae*: 2 μg/mL *E. coli*: 16 μg/mL | *Jupudi, Azam & Wadhwani (2019)* |
| 10 | MurD | *S. aureus*: 13.37 μM | *S. aureus*:32 μg/mL *K. pneumoniae*: 4 μg/mL *E. coli*: 16 μg/mL | *Jupudi, Azam & Wadhwani (2019)* |
| 11 | MurD | *E. coli*: 85 μM | Not determined | *Humljan et al. (2008)* |
| 12 | MurD | *E. coli*: 8.4 μM | No antimicrobial activity | *Sosič et al. (2011)* |
| 13 | MurE | *M. tuberculosis*: 57 μM | *M. tuberculosis*: 4 μg/mL *M. bovis* BCG: 5 μg/mL | *Guzman et al. (2010)* |
| 14 | MurE | *M. tuberculosis*: 36 μM | *M. tuberculosis*: 10 μg/mL *M. smegmatis*: 10 μg/mL *S. aureus*: 0.5 μg/mL | *Guzman et al. (2011)* |
| 15 | MurF | *S. pneumoniae*: 8 μM | Not determined | *Gu et al. (2004)* |
| 16 | MurF | *S. pneumoniae*: 1 μM | Not determined | |
| 17 | MurF | *S. pneumoniae*: 22 nM | Not determined | *Stamper et al. (2006)* |
| 18 | MurF | *S. pneumoniae*: 0.42 μM *E. coli*: 81 μM *S. aureus*: 91 μM | *S. pneumoniae*: 16–32 μg/mL | *Hrast et al. (2013)* |
| 19 | MurF | *E. coli*: 26 μM | LPS-defective *E. coli*: 8–16 μg/mL *S. aureus*: 8–16 μg/mL *E. faecalis*: 8–16 μg/mL *E. faecium*: 8-16 μg/mL | *Baum et al. (2007)* |
| 20 | MurF | *E. coli*: 24 μM | *E. coli*: 8 μg/mL *S. aureus*: >32 μg/mL | *Baum et al. (2009)* |
| 21 | MurF | *E. coli*: 29 μM | *E. coli*: 8 μg/mL *S. aureus*: 32 μg/mL | *Baum et al. (2009)* |
| 22 | MurD MurE MurF | *E. coli* MurD: 2 μM *S. aureus* MurE: 6 μM *E. coli* MurF: 2 μM | *E. coli*: >128 μg/mL P. aeruginosa: 128 μg/mL *E. faecalis*: >128 μg/mL *S. aureus*: >128 μg/mL | *Tomašić et al. (2010)* |

**Table 1** (*continued*)

| ID | Target | IC$_{50}$ | MIC | Reference |
|---|---|---|---|---|
| 23 | MurD<br>MurE | *E. coli* MurD: 690 μM<br>*E. coli* MurE: 89 μM | *E. faecalis: 128 μg/mL*<br>*S. aureus*: No activity<br>*H. influenzae*: No activity | *Perdih et al. (2014)* |
| 24 | MurC<br>MurD<br>MurE<br>MurF | *E. coli* MurC: 94 μM<br>*E. coli* MurD: 92 μM<br>*E. coli* MurE: 32 μM<br>*E. coli* MurF: 12 μM | *S. aureus*: 256 μg/mL<br>*E. faecalis*: >256 μg/mL<br>*H. influenzae*: 256 μg/mL | *Perdih et al. (2015)* |
| 25 | MurC<br>MurD<br>MurE<br>MurF | *E. coli* MurC: 41 μM<br>*E. coli* MurD: 60 μM<br>*E. coli* MurE: 93 μM<br>*E. coli* MurF: 89 μM | Not determined | *Perdih et al. (2014)* |
| 26 | MurD<br>MurE | *E. coli* MurD: 8.2 μM<br>*S. aureus* MurD: 6.4 μM<br>*E. coli* MurE: 180 μM<br>*S. aureus* MurE: 17 μM | *S. aureus*: 8 μg/mL<br>*E. faecalis*: 64 μg/mL<br>*E. coli*: >128 μg/mL | *Tomašić et al. (2012a)*;<br>*Tomašić et al. (2012b)* |
| 27 | MurX | 550 nM | *M. smegmatis*: 6.25 μg/mL<br>*M. avium*: <0.063 μg/mL<br>*M. intracellulare*: <0.063 μg/mL | *Hotoda et al. (2003a)*;<br>*Hotoda et al. (2003b)* |
| 28 | MurX | *M. tuberculosis*: 41 nM | *M. tuberculosis*: 0.037 μM | *Tran et al. (2017)* |
| 29 | Transpe-ptidase | Not determined | *M. tuberculosis*: 0.1 μg/mL<br>*M. abscessus*: 0.4 μg/mL | *Gupta et al. (2021)* |

high level of homology between the enzymes involved (*Glauner, Höltje & Schwarz, 1988*). The biosynthesis of peptidoglycan can be spatially separated into three stages; cytoplasmic stage, which is responsible for the biosynthesis of peptidoglycan precursors, membrane-associated stage, which involves attachment of precursors to the inner membrane, and ultimately extracellular stage, where the polymerisation and crosslinking occurs (*Raghavendra, Patil & Mukherjee, 2018*).

The cytoplasmic stage of peptidoglycan synthesis begins with the conversion of fructose-6-phosphate to Uridine-diphosphate-GlcNAc (UDP-GlcNAc), requiring the activity of three enzymes, GlmS, GlmM, and GlmU (*Van Heijenoort, 2001*). GlmS, an aminotransferase, is responsible for the conversion of fructose-6-phosphate into glucosamine-6-phosphate (*Durand et al., 2008*). Subsequent conversion to glucosamine-1-phosphate is catalysed by GlmM, a mutase (*Li et al., 2012*). The acetylation and uridylation of glucosamine-1-phosphate is conducted by the bifunctional enzyme, GlmU, which completes the synthesis of UDP-GlcNAc (*Zhang et al., 2009*). The conversion of UDP-GlcNAc to UDP-MurNAc subsequently takes place due to the activities of the enolpyruvyl transferase and the enolpyruvyl reductase, MurA and MurB respectively. MurA transfers the enolpyruvyl moiety to UDP-GlcNAc, followed by the reduction catalysed by MurB, which converts the enolpyruvyl moiety to D-lacoyl, yielding UDP-MurNAc (*Benson et al., 1993*; *Jankute et al., 2015*; *Xu et al., 2014*). Thereafter, the assembly of the pentapeptide side chain is facilitated by sequential activities of functionally similar ATP-dependent ligases, MurC, MurD, MurE, and MurF (*Barreteau et al., 2008*) in parallel with hydrolysis of ATP (Fig. 2), resulting in the addition of L-Ala, D-Glu, mDAP and D-Ala-D-Ala, respectively,

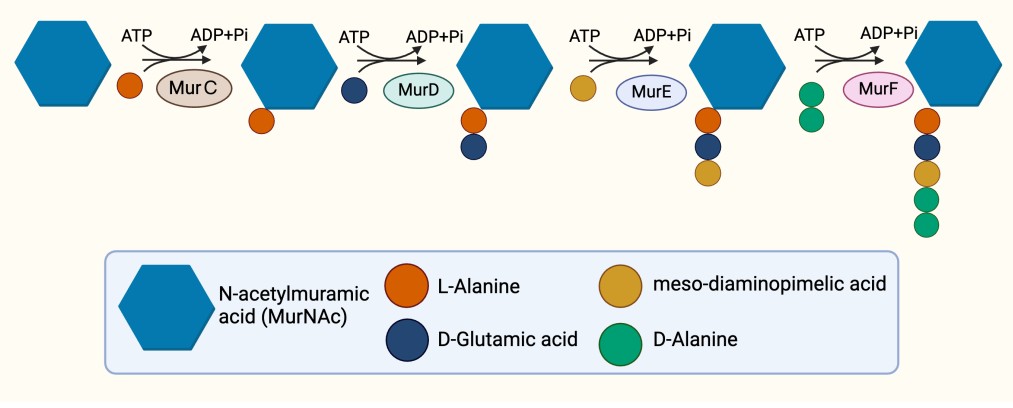

**Figure 2** The cytoplasmic synthesis of Park's nucleotide.

and ultimately resulting in the generation of Park's nucleotide (UDP-MurNAc-L-ala-D-Glu-mDAP-D-Ala-D-Ala) (*Munshi et al., 2013*).

Following the generation of Park's nucleotide (UDP-MurNAc-pentapeptide), the association with cytoplasmic membrane is facilitated by MurX/MraY, creating a membrane-bound peptidoglycan precursor (Lipid I) (*Ben et al., 2013*). Next, GlcNAc is attached to Lipid I by MurG, resulting in Lipid II, the monomeric units of peptidoglycan (*Jha et al., 2012*). It is generally believed that Lipid II is flipped to the periplasm by a "flippase" called MurJ, but the detailed mechanism remains unexplored (*Sanduo et al., 2018*; *Sham et al., 2014*). Thereafter, polymerisation and crosslinking of the monomeric units occur in the periplasm in the last stage of peptidoglycan synthesis, which involves transglycosylation and transpeptidation of the monomeric units. Bifunctional enzymes, PonA1 and PonA2 are associated with polymerisation and crosslinking of monomeric units. The glycosyltransferase domain of PonA1 and PonA2 achieve polymerisation by ligating the GlcNAc moiety with muranyl moiety while the transpeptidase domain contributes to the formation of crosslinking between mDAP and D-Ala (3 → 4 linkage) of the adjacent pentapeptide side chain, yielding mature peptidoglycan (Fig. 3) (*Hett, Chao & Rubin, 2010*; *Sauvage et al., 2008*). Ultimately, the biosynthesis of peptidoglycan is completed after anchoring to arabinogalactan by Lcp1 phosphotransferase (*Harrison et al., 2016*).

## Mur ligase inhibitors for combating TB

Cell wall peptidoglycan biosynthesis has been heavily investigated and exploited as an antibiotic target since the discovery of penicillin. However, both ß-lactam and glycopeptides are ineffective against *M. tuberculosis* due to robust ß-lactamases and the intrinsic resistance imparted by the *M. tuberculosis* cell wall (*Chambers et al., 1995*; *Li, Zhang & Nikaido, 2004*; *Nikaido, 2001*). Frontline TB drugs established the viability of targeting cell wall in TB treatment such as isoniazid targets mycolic acid synthesis, ethambutol targets arabinogalactan synthesis, and D-cycloserine targets D-Ala:D-Ala ligase (*Prosser & De Carvalho, 2013*; *Timmins & Deretic, 2006*; *Zhu et al., 2018*). However, peptidoglycan

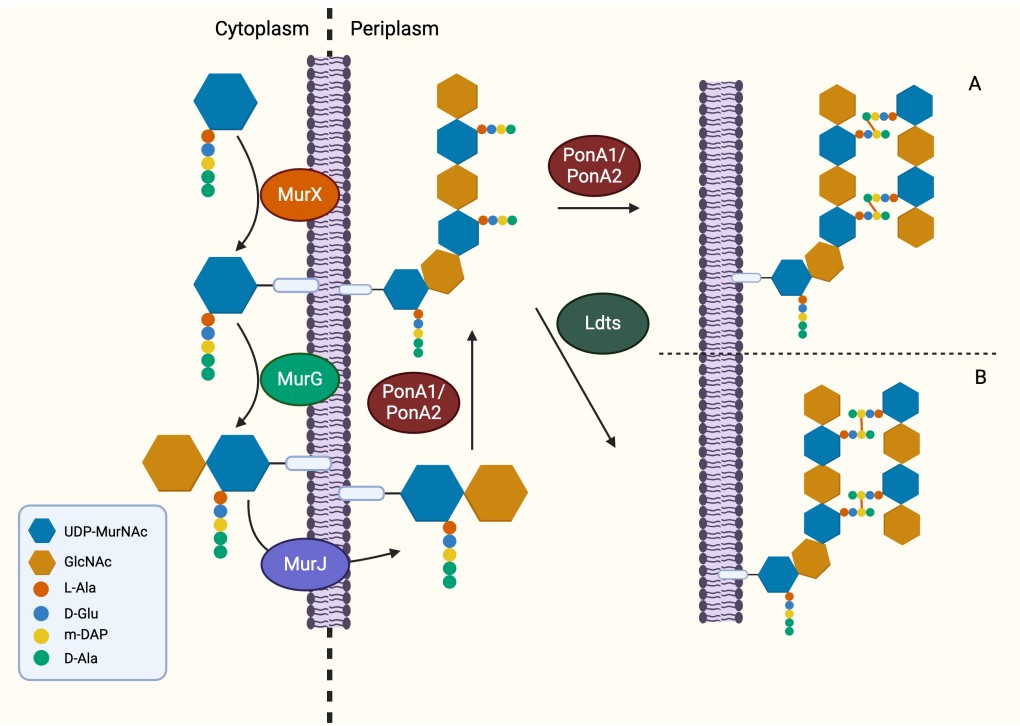

**Figure 3** **The membrane-associated stage and the extracellular stage of peptidoglycan synthesis.** Two distinctive linkage can be facilitated by different enzymes in transpeptidation, the ultimate step of peptidoglycan synthesis. (A) The transpeptidation domain of PonA1/PonA2 facilitate the formation of 3-4 linkage and (B) Ldt$_{MT1-5}$ facilitate formation of 3-3 linkage which is commonly seen in dormant or non-replicating Mtb.

inhibitors approved for TB therapy are limited. According to 2022 guidelines, ß-lactam-based antibiotic, imipenem-cilastatin or meropenem, can be used in the treatment of MDR or XDR-TB cases. However, due to the lack of clinical evidence, they are conditionally recommended and only administered when frontline and second-line medication cannot be used (*World Health Organization, 2022*). Recent clinical trials on meropenem also demonstrated minimal improvement in treatment outcomes (*De Jager et al., 2022*). The D, D-transpeptidases responsible for the formation of typical $3 \rightarrow 4$ crosslinking bonds are the essential target for ß-lactam antibiotics (*Ghuysen, 1991*). Previous research indicated that an exceptionally high percentage (80%) of atypical $3 \rightarrow 3$ linkage are formed by L, D-transpeptidases in dormant *M. tuberculosis* during non-replicating persistence (Fig. 1) (*Lavollay et al., 2008*). Thus, sole inhibition of D, D-transpeptidases by ß-lactam antibiotics is insufficient in clearing a mycobacterial infection (*Gupta et al., 2010*). Recently, carbapenems which target both D, D-transpeptidase and L, D-transpeptidase have been used in treating MDR-TB and XDR-TB (*Kumar et al., 2019*; *Van Rijn et al., 2019*).

There is currently great interest in developing novel anti-tubercular drugs targeting the early stages of peptidoglycan synthesis with Mur ligases being of particular interest. There are good reasons for the interest here, including; the Mur ligases are present in all clinically important bacterial pathogens but absent in mammals, providing potential

board-spectrum inhibition and selectivity (*Kouidmi, Levesque & Paradis-Bleau, 2014*), and; the conserved structure of binding sites among Mur ligases can be exploited by designing inhibitors with potential multitarget inhibition which target multiple Mur ligases. The probability of resistance against multitarget inhibitors reduces exponentially because at least two advantageous mutations are required in the same generation to develop resistance. Hence, multitarget inhibitors can be clinically beneficial in difficult-to-treat bacteria such as *M. tuberculosis* (*Silver, 2007*).

The only published protein structure of Mur ligases from *M. tuberculosis* remains MurE (*Basavannacharya et al., 2010*); nonetheless, the X-ray crystallography of Mur ligases from different species has been solved, aiding rational drug design. Homology models of Mur ligases for *M. tuberculosis* were constructed based on the amino acid sequence similarity with these corresponding homologues. The crystal structure of MurC from *Haemophilus influenzae* was utilised as a template for homology modelling of *M. tuberculosis* MurC, while *Escherichia coli* was exploited for MurD modelling and *Streptococcus pneumoniae* was exploited for MurF modelling (*Kotnik et al., 2007*; *Mol et al., 2003*; *Moraes et al., 2015*; *Yan et al., 2000*). Since good structural homology is observed, it is expected that inhibitors of Mur ligases of other species might inhibit mycobacterial Mur ligases as well. Herein, we review the inhibitors of Mur ligases from various bacterial species which might serve as a good starting point in developing mycobacterial Mur ligase inhibitors.

## Inhibitors of MurC

MurC facilitates the initial step of the pentapeptide side chain assembly and is the preliminary step in the cytoplasmic stage of peptidoglycan synthesis. Enormous effort has been devoted to the discovery and development of molecules which interfere with this catalytic process (Fig. 4). Many published MurC inhibitors were transition state analogues such as phosphinate and Banzofuran acyl-sulfonamide.

Initially, a series of phosphinate compounds were synthesised as transition state analogue inhibitors for MurC. The phosphinate inhibitors demonstrated significant potency against MurC with the most potent compound exhibiting an $IC_{50}$ of 42 nM (Fig. 4, compound 1) (*Marmor et al., 2001*; *Reck et al., 2001*). Due to the structural similarity of Mur ligases, phosphinates were found to be capable of inhibiting MurD-F, which will be discussed in the following sections.

Benzofuran acyl-sulfonamides was identified as an inhibitor of MurC with an $IC_{50}$ value of 2.3 μM (Fig. 4, compound 2). However, it displayed significant protein binding liabilities which led to notable reduction in MurC inhibition (*Ehmann et al., 2004*).

Besides transition state analogues, phosphate or diphosphate mimetics are commonly seen in Mur ligase inhibitors (*Maitra et al., 2019*). These compounds generally incorporate a heterocyclic moiety such as rhodanine or thiazodindione. Benzylidene rhodanines (Fig. 4, compound 3) were identified as MurC inhibitors, with many exhibiting low micro-molar inhibition ($IC_{50}$ = 12–24 μM). The whole-cell inhibitory activity of benzylidene rhodanine was also evaluated, with negligible antibacterial activity displayed for *E. coli* (MIC > 65 μM) and modest activity observed in methicillin-resistant *Streptococcus aureus* (MIC = 31–93 μM) (*Sim et al., 2002*). Two ATP-competitive inhibitors (Fig. 4,

**Figure 4** The catalytic mechanism of MurC with inhibitors and corresponding IC$_{50}$ value.

compound 4 and 5) were discovered *via* high-throughput screening (HTS). The potency of compound 4 (a quinoxaline) was evaluated against MurC isolated from six microorganisms (*E. coli*, *Klebsiella pneumoniae*, *Proteus mirabilis*, *Haemophilus influenzae*, *Acinetobacter baylyi*, *Pseudomonas aeruginosa*). Interestingly it only inhibit MurC isolated from Enterobacteriaceae (IC$_{50 E.coli}$ = 30.2 µM, IC$_{50 K.pneumoniae}$ = 26.4 µM, IC$_{50 P.mirabilis}$ = 41.4 µM) but failed to inhibit MurC from other bacterial families. This suggests that there are distinct structural features in Enterobacterial MurC which are required for the inhibition by the quinoxaline (*Zawadzke et al., 2008*). Nevertheless, compound 4 was not antibacterial even with highly drug-susceptible, efflux-defective *E. coli* strain (*E. coli* ΔTO1C) (*Zawadzke et al., 2008*).

Pyrazolopyrimidine (compound 5) inhibited MurC isolated from *E. coli* and *P. aeruginosa* at nanomolar levels (IC$_{50}$ of 12 nM and <10 nM respectively). In contrast to quinoxaline, pyrazolopyrimidines displayed antibacterial activity in both TO1C-mutated *E. coli* and efflux-deficient *P. aeruginosa* (MIC$_{E.coli\Delta TO1C}$ = 3.13 µM, MIC$_{P.aeruginosa}$ 546 = 1.56 µM) (*Hameed et al., 2014*; *Humnabadkar et al., 2014*). Nevertheless, no antibacterial activity was observed in wild-type *E. coli* and *P. aeruginosa*, suggesting the low permeability and high efflux rate restrain the development of pyrazolopyrimidine as an antibacterial

agent and was not further tested on *M. tuberculosis* (*Hameed et al., 2014*; *Humnabadkar et al., 2014*).

Apart from chemical entities, short peptide ligands were also examined as MurC inhibitors. Two peptide MurC inhibitors (N-Asp–His–Arg–Asn–Pro–Asn–Tyr–Ser–Trp–Leu–Lys–Ser-COOH and N-Cys–Gln–Asp–Thr–Pro–Tyr–Arg–Asn–Cys-COOH) was discovered *via* phage display, but $IC_{50}$ values of 1.5 mM and 0.9 mM respectively were disappointing (*El Zoeiby et al., 2003*). Feglymycins of *Streptomyces*, a 13-mer peptide antibiotic was also revealed with exceptional capability to inhibit both MurA and MurC of *E. coli* ($IC_{50}$ = 3.4 µM and 0.3µM, respectively) and *S. aureus* ($IC_{50}$ = 3.5 µM and 1.0 µM, respectively) (*Hänchen et al., 2013*). Additionally, feglymycin demonstrated antibacterial activity on three drug-resistant *S. aureus* strains with MICs between 0.25 µM to 1 µM, highlighting the potential of MurC as a novel pathway in combating rapid-emerging resistance and warranting further consideration (*Hänchen et al., 2013*).

## Inhibitors of MurD

MurD is responsible for the subsequent ligation of D-Glu onto L-Ala, which results in the formation of UDP-MurNAc-L-Ala-D-Glu. It is known that MurD possesses high stereospecificity toward D-Glu. The binding sites for D-Glu are conserved across almost all bacteria, which made MurD an ideal target for antibiotic development. A significant amount of effort has been dedicated to finding MurD inhibitors (Fig. 5). In 2003, a macrocyclic MurD inhibitor (Fig. 5, compound 6) was discovered *via* a structure-based rational design. The synthesised macrocyclic inhibitors demonstrated inhibition against MurD at the low micromolar range ($IC_{50}$ range between 0.7−9.0 µM) (*Horton et al., 2003*). However, no further progress was documented in the development of this molecule which might be due to the lack of any reported antibacterial activity.

More recently, virtual screening has been heavily deployed due to time and cost efficiency in identifying inhibitory scaffolds, which can then be subsequently optimised to improve potency and antibacterial activity. A lead compound containing D-Glu moiety was discovered as a MurD inhibitor *via* virtual screening, exploiting the high specificity towards D-Glu. Preliminary structural optimisation of the D-Glu-based scaffold showed that a compound containing a 5-benzylidenerhodanine can inhibit MurD in micromolar concentration ($IC_{50}$ = 174–206 µM) (*Tomašić et al., 2009*). Further optimisation was conducted and led to the discovery of 5-benzylidenerhodanine and 5-Benzylidenethiazolidine-2,4-dione, inhibitors with improved potency ($IC_{50}$ = 85 µM and 45 µM, respectively) (*Zidar et al., 2010*). The binding mode of 5-Benzylidenerhodanine was evaluated through X-ray crystallography and revealed the compound binds to the active site of MurD, suggesting the compounds are the ideal starting point of developing a MurD-targeting antibiotic (*Zidar et al., 2010*). In another study, the lead compound (Fig. 6, compound 7) was developed by combining 2-thioxothiazolidin-one-ring, a uracil mimicry that binds to MurD, and glutamic acid and demonstrated good potency against MurD with $IC_{50}$ of 10 µM (*Tomašić et al., 2011*). Further exploration of the structure–activity relationship (SAR) on the scaffold demonstrated three novel 5-benzylidenethiazolidin-4-one inhibitors that can inhibit MurD with $IC_{50}$ less than 50 µM (*Zidar et al., 2011*).

**Figure 5** The catalytic mechanism of MurD with inhibitors and corresponding $IC_{50}$ value.

Unfortunately, these compounds lacked antibacterial activity, with only one compound (Fig. 6, compound 8) displaying poor antibacterial activity against Gram-positive *S. aureus* and *E. feacalis* with MIC of 128 µg/mL (*Zidar et al., 2011*).

In 2019, a virtual screening based on *S. aureus* MurD structure followed by docking energy and molecular dynamics (MD) simulation also successfully identified another scaffold. The following *in vitro* data indicated that the identified hit displayed significant inhibitory activity ($IC_{50} = 7$ µM) and mild antibacterial activity (MIC $= 128$ µg/mL) against *S. aureus* and *Bacillus subtilis*, suggesting it was a promising candidate for development (*Azam & Jupudi, 2019*). Subsequent structural modifications were performed on the scaffold to improve the affinity towards MurD. The modified compounds demonstrated inhibitory activity with $IC_{50}$ ranged from 13.37–79.27 µM. Among the modified molecules, the benzothiazol-2-ylcarbamodithioate substitutes, compound 9 and 10 (Fig. 5), possess the greatest inhibitory property against *S. aureus* with $IC_{50}$ of 22.33 and 13.37 µM respectively. Furthermore, unlike most of the Mur ligases inhibitors, compound 13 and 14 exhibited significant bactericidal activity against multiple microorganisms ($MIC_{S.aureus} = 2$ and 32 µg/mL, respectively. $MIC_{K.pneumoniae} = 2$ and 4 µg/mL, respectively. $MIC_{E.coli} = 16$ and 16 µg/mL, respectively) compared to the current standard treatment ciprofloxacin and gentamicin ($MIC_{S.aureus} = 2$ and 8 µg/mL, respectively) (*Jupudi, Azam & Wadhwani,*

**Figure 6** The catalytic mechanism of MurE with inhibitors and corresponding $IC_{50}$ value.

*2019*). The promising antibacterial activity in a broad spectrum of bacteria warrants further investigation and requires validation on *M. tuberculosis*.

Another compound, naphthalene-N-sulfonyl-d-Glu, was designed to mimic the tetrahedral transition state of MurD. The inhibitors exhibited the competitive inhibition mode to D-Glu with the $IC_{50}$ value of 280 µM. The structure of inhibitor-MurD complex also demonstrated that the inhibitor binds to the enzyme active site *via* hydrogen bond and lipophilic interaction formed between D-Glu moiety and D-Glu-binding pocket (*Kotnik et al., 2007*). The subsequent structural optimisation fully elucidated that the D-Glu moiety and sulphonamide bond are essential for MurD inhibition as D-Glu provides specificity and sulphonamide bond provides geometric similarity with the tetrahedral intermediate (*Obreza & Gobec, 2004*). The structural optimisation demonstrated that introducing arylalkyloxy substituent on position 6 of the naphthalene further improved the inhibition, with compound 11 (Fig. 5) being the most potent compound in this study with $IC_{50}$ at 85 µM (*Humljan et al., 2008*). The superior potency was backed by a free binding energy calculation, indicating the incorporation of 4-cyanophenyl of compound 11 contributes to the most potent binding compared to other analogues (*Perdih, Bren & Solmajer, 2009*).

The naphthalene-N-sulfonyl-D-Glu inhibitors were further developed into second-generation sulphonamide inhibitors by rigidising the D-Glu moiety. The resulting compound 12 (Fig. 5) exhibited a further 10-fold improvement in inhibition activity with

the $IC_{50}$ of 8.4 µM compared to the parent compound 11 (*Sosič et al., 2011*). The recent computational analysis of second-generation sulphonamide inhibitors also elucidated that the rigid D-Glu moiety allows them to be more favourable in binding with the calculated binding free energy of −3.4 kcal/mol, compared to unrigidised inhibitor, with the binding value of −2.6 kcal/mol and further attention is warranted in this space (*Perdih, Wolber & Solmajer, 2013*). Unfortunately, these compounds were found to be ineffective in inhibiting bacterial growth, which might be attributed to poor penetration into bacterial cell walls. Despite the lack of antibacterial activity on the species tested, the enzyme potency and specificity demonstrated suggests an opportunity for further development as anti-TB agents (*Sosič et al., 2011*).

### Inhibitors of MurE

MurE facilitates the attachment of m-DAP onto D-Glu moiety of UDP-MurNAc-dipeptide, yielding UDP-MurNAc-tripeptide. Interestingly, despite there beening fewer MurE inhibitors published, many of the MurE inhibitor demonstrated inhibition of the *M. tuberculosis* MurE, as well as antitubercular activity. Previous studies have shown that analogues of m-DAP and tetrahedral transition state (phosphinates) both possess potency against MurE (*Auger et al., 1996*; *Strancar et al., 2007*). Another scaffold, 3-methoxynordomesticine, which was isolated from the Colombian plant *Ocotea macrophylla*, was identified with anti-tubercular properties. The most potent compound (Fig. 6, Compound 13) exhibited MurE-inhibiting activity $IC_{50}$ of 57 µM and anti-tubercular activity with MIC <5 µg/mL against *M. tuberculosis* H37Rv. Further structural optimisation of 3-methoxynordomesticine is suggested to reduce cytotoxicity while improving affinity for MurE (*Guzman et al., 2010*). Similar to 3-methoxynordomesticine, (S)-Leucoxine isolated from the Colombian Lauraceae tree *Rhodostemonodaphne crenaticupula* was also found to hinder the growth of *M. tuberculosis*. An explorative SAR study was conducted focused on the common tetrahydroisoquinoline scaffold between 3-methoxynordomesticine and (S)-Leucoxine. After three rounds of optimisations, three tetrahydroisoquinolines were able to inhibit *M. tuberculosis* MurE with $IC_{50}$ < 111 µM, with antimycobacterial activity displayed (MIC = 20–60 µg/mL), conferring tetrahydroisoquinolines as MurE-targeting anti-tubercular agents (*Guzman et al., 2015*).

An alternative scaffold identified, N-methyl-2-alkenyl-4-quinolone, was isolated from *Euodia rutaecarpa*, a Chinese medical plant and display antimycobacterial activity with MIC of 1 µg/mL against *Mycobacterium smegmatis* (*Wube et al., 2011*). A subsequent study elucidated the antimycobacterial property of N-methyl-2-alkenyl-4-quinolones (Fig. 6, compound 14) by demonstrating inhibition against *M. tuberculosis* MurE ($IC_{50}$ = 36–72 µM). Docking simulation revealed that quinolones interact to the pocket near N-terminus uracil recognition site of MurE with a competitive inhibition pattern against UDP-MurNAc-dipeptide substrate observed (*Guzman et al., 2011*).

### Inhibitors of MurF

MurF is responsible for the ligation of D-ala-D-ala dipeptide during cytoplasmic peptidoglycan synthesis, which is also the ultimate step in Mur ligases cascade. Significant
numbers of MurF inhibitors have been investigated. Two MurF-inhibiting leads (Fig. 7, compound 15 & 16) (IC$_{50}$ = 8 μM and 1 μM, respectively) were reported *via* affinity selection screening by Abbott lab, followed by several rounds of structure-based optimization (*Gu et al., 2004*). Guided by the crystal structures reported, the optimisation successfully improved the potency by 40-fold, improving the IC$_{50}$ from 1 μM to 22 nM (Fig. 7, compound 17), albeit the improved compound still lacked bactericidal activity (*Gu et al., 2004*; *Stamper et al., 2006*). The mechanism of action of compound 17 was further elucidated by the following co-crystallisation study, which shows that compound 17 occupies the uridine-ribose binding pocket on the N-terminal domain and induces conformational closure (*Longenecker et al., 2005*). Even though the compound was never adapted commercially due to the absence of antibacterial activity, this study established a profound insight into the interaction between the compound and MurF. More recently, two SAR studies designed and synthesised a series of cyanothiophene MurF inhibitors by introducing structural modifications on compound 15, followed by potency examination on MurF from three species (*S. pneumoniae*, *E. coli* and *S. aureus*) and MIC determination (*Hrast et al., 2014*; *Hrast et al., 2013*). Ultimately, the modified compounds (Fig. 7, compound 18) successfully enhanced the inhibition activity to nanomolar level towards MurF of *S. pneumoniae*, yet only poor inhibition was demonstrated in *E. coli* and *S. aureus* (IC$_{50}$ <100 μM) (*Hrast et al., 2013*). Co-crystallisation demonstrated that the newly developed cyanothiophene-based inhibitors interact with MurF in a similar fashion to the parent compound which induced interdomain closure, and several modified compounds display moderate bactericidal activity against *S. pneumoniae* and *E. coli* (MIC = 16–32 μg/mL) (*Gu et al., 2004*; *Hrast et al., 2014*; *Hrast et al., 2013*). Moreover, multiple computational approaches were exploited in validating and optimising the compound-enzyme interaction of compound 15 and compound 16 as well as design new MurF inhibitors based on the scaffold, which now require further *in vivo* examination (*Azam & Jupudi, 2017*; *Khedkar, Malde & Coutinho, 2007*; *Taha et al., 2008*).

A series of MurF-inhibiting compounds, 8-hydroxyquinoline, was also determined with significant potency (IC$_{50}$ = 0.33–25 μM) and moderate bactericidal activity (MIC = 8–32 μg/mL) (*Baum et al., 2007*). However, the subsequent assays showed that the antibacterial activity of 8-hydroxyquinoline was predominantly contributed by its well-known chelating property (*Fraser & Creanor, 1975*; *Rohde et al., 1977*). To circumvent this issue, another compound, 4-phenylpiperidine (Fig. 7, compound 19), discovered *via* 8-hydroxyquinoline-based pharmacophore modelling, showed bactericidal activity *via* MurF inhibition (IC$_{50}$ = 26 μM) in LPS-modified *E. coli* (MIC = 8 μg/mL) but absent in wild type *E. coli* (MIC > 32 μg/mL) due to poor cell penetration (*Baum et al., 2007*). The pharmacophore model also led to the discovery of two diarylquinolines (Fig. 7, compound 20 and 21) which displayed MurF-inhibiting properties with IC$_{50}$ of 24 and 29 μM, respectively. Moreover, compound 20 and 21 demonstrated antibacterial activity in both Gram-positive and Gram-negative species (MIC = 2–8 μg/mL) with accumulated MurF reactants and decreased MurF products, suggesting that the antibacterial activity of diarylquinolines was induced by MurF inhibition (*Baum et al., 2009*). This finding

**Figure 7** The catalytic mechanism of MurF with inhibitors and corresponding IC$_{50}$ value.

warranted the development of a MurF inhibitor into novel anti-TB chemotherapy, with diarylquinolines showing promising results in a broad spectrum of bacteria.

## Multitarget inhibitors

It stands to reason that multitarget inhibitors may represent a therapeutic benefit over single-target inhibitors in certain conditions (*Silver, 2007*). The threshold for potency may decrease with the synergistic inhibition of multiple pathway enzymes, reducing the costs associated with treatment and the risk of off target effects. Further, the development of resistance is less likely to occur since it would require simultaneous mutations on two or more targeted genes in a single generation (*Škedelj et al., 2011*). Mur ligases are ideal targets for designing multitarget inhibitors due to the similar structures and catalytic mechanisms among Mur ligases family (*Bouhss et al., 1999*). The highly conserved residues in ATP-binding sites and UDP-substrate-binding sites of Mur also underline the viability of designing compounds that interact and potentially inhibit multiple ligases simultaneously (*El Zoeiby et al., 2003*). The published multitarget Mur inhibitors can be generally categorised into three main types: (A) competitive ATP inhibition, (B) Inhibitor mimicking UDP-MurNAc moiety of UDP substrate, and (C) compounds which mimic the common tetrahedral intermediate of Mur ligases (*Tomašić et al., 2010*).
## Targeting ATP-binding site

ATP-binding pocket of Mur ligases, which was identified as a highly conserved region among the Mur ligases family, were investigated as candidate sites for multitarget inhibitors (*Bouhss et al., 1999*; *Bouhss et al., 1997*). Different strategies were deployed to identify competitive inhibitors that target ATP-binding on Mur ligases (*Šink et al., 2008*). Structure-based virtual screenings were exploited which led to the discovery of several compounds that exhibited modest potency against MurC and MurD by competitive inhibition. Those compounds act as the starting points for the design of multiple Mur ligase inhibitors targeting the ATP-binding site (*Tomašić et al., 2012a*; *Tomašić et al., 2012b*). Additionally, N-Acylhydrazones was identified as a multitarget Mur inhibitor *via* compound library screening. During optimisation, the potency against MurC and MurD was observed on N-acylhydrazone bearing 2,3,4-trihydroxyphenyl group, with $IC_{50}$ of 123 µM and 230 µM respectively. Results also showed that the inhibitor predominantly occupies the ATP-binding site with partial binding to UDP-binding pocket, leads to the simultaneous inhibition of MurC and MurD (*Šink et al., 2008*).

## UDP-substrate binding

UDP-substrate-binding pockets are another highly conserved region in Mur ligases since the UDP-moiety is present throughout the entire Mur ligase cascade. The 5-benzylidenethiazolidin-4-ones analogue (Fig. 8, compound 22) which incorporated rhodanine as a diphosphate surrogate and the 2,3,4-hydroxyphenoyl group as hydrogen-bond-forming moiety was designed to bind within the ATP- or UDP-binding site. The design of the compound was successful, with the resulting compound potency against MurD-F in the low micromolar range ($IC_{50} = 2–6$ µM). Kinetic analysis revealed that the inhibition of 5-benzylidenethiazolidin-4-ones was contributed solely by the interaction with UDP-binding pocket. Unfortunately, 5-benzylidenethiazolidin-4-ones was devoid of antibacterial activity (*Tomašić et al., 2010*). Surprisingly, the analogues of UDP-MurNAc which were expected to induce multiple Mur ligases inhibition only demonstrated potency against MurE of *M. tuberculosis*, indicating further optimisation is required for the scaffold (*Hervin et al., 2020*).

## Tetrahedral intermediate mimics

A frequently exploited strategy in design of MurD inhibitors is mimicking MurD tetrahedral intermediate, which is commonly achieved by phosphinate or N-sulfonyl derivatives (*Kotnik et al., 2007*; *Obreza & Gobec, 2004*). The discovery of the peptide-sulfonamides bearing D-Glu group which inhibit both MurD and MurE have shown that product-like MurD inhibitors can serves as a substrate-like inhibitor for MurE with noteworthy potency displayed in both enzymes (*Humljan et al., 2006*). Moreover, it was observed by structural comparison that the MurD residues responsible for binding of D-Glu also present in MurE active site (*Gordon et al., 2001*). These results accentuate the possibility of designing dual (MurD/MurE) inhibitor *via* MurD tetrahedral intermediate mimicry.

**Figure 8  Multi-target inhibitors of Mur ligases.**

## Arresting the catalytic activity

To search for D-Glu-based dual inhibitors, a pharmacophore-based virtual screening followed by potency assessment revealed that benzene 1,3-dicarboxylate moiety serves as a rigid replacement for D-Glu and led to greater potency against both MurD and MurE (*Perdih, Bren & Solmajer, 2009*). Subsequently, those discovered leads, benzene-1,3 dicarboxylate bearing 2,5-dimethylpyrrole (Fig. 8, compound 23) and benzene-1,3-dicarboxylate bearing furane moiety (Fig. 8, compound 24) were selected for further optimisation. The optimisation on compound 23 demonstrated that incorporation of the rhodanine moiety (compound 25) resulted in enhanced potency and expansion of the inhibition to all four Mur ligases (MurC-F) (*Perdih et al., 2014*). Also the optimisation of compound 24 showed that it exhibited inhibitory property in all four Mur ligases (MurC-F) in low micromolar level (*Perdih et al., 2015*). Interestingly, these two similar scaffolds displayed different inhibitory mode against MurD substrates. Benzene-1,3 dicarboxylate 2,5-dimethylpyrrole (Fig. 8, compound 23) exhibited competitive inhibition to D-Glu while partial uncompetitive and partial non-competitive inhibition to ATP and UMA respectively. Alternatively, furan-based benzene 1,3-dicarboxylic acid (Fig. 8, compound 24) display apparent competitive inhibition to ATP (*Perdih et al., 2014*; *Perdih et al., 2015*). Hence it appears that Benzene-1,3 dicarboxylate 2,5-dimethylpyrrole fall into UDP-substrate mimicry inhibitor category while furan-based benzene 1,3-dicarboxylic acid act as an ATP-competitive inhibitor. Unfortunately, antibacterial property was devoid in all benzene 1,3-dicarboxylic acid analogues, with only one furan-based compound showing modest activity against *S. aureus* (MIC = 32 μg/mL) (*Perdih et al., 2015*). Similarly, another D-Glu based multitarget inhibitor was developed *via* modification of previously

identified MurD inhibitor. The resulting compound (Fig. 8, compound 26) exhibited dual inhibition against both MurD and MurE ($IC_{50}$ = 6.4 and 17 µM, respectively) and bactericidal activity against *S. aureus* (MIC = 8 µg/mL) (*Tomašić et al., 2012a*; *Tomašić et al., 2012b*). The following research validates the formation of inhibitor-MurE complex, which is mainly contributed by hydrogen bond formed with central domain and docking of D-Glu and thiazolidine-4-one ring towards UDP-MurNAc-tripeptide-binding pocket and uracil-binding pocket respectively (*Azam & Jupudi, 2019*). To summarise, despite the lack of antibacterial activity in species tested, the development of multitarget chemotherapy is of great interest for species like *M. tuberculosis* where slow growth characteristics may represent entirely different outcomes.

## Inhibitors of MurX

Following the sequential cytoplasmic synthesis of UDP-MurNAc-pentapeptide, the next stage of peptidoglycan synthesis, known as membrane-associated stage, facilitates the translocation of this peptidoglycan precursor.

The UDP-MurNAc-pentapeptide is transported to the periplasm *via* three crucial enzymes MurX, MurG and MurJ. Likewise, the essentiality and specificity of these enzymes make them the ideal target for antibiotics. Hence, besides exploiting Mur ligases due to their structural resemblance, work has also focussed on targeting essential enzymes involved in the membrane-associated stage of peptidoglycan synthesis. Capuramycin, a nucleoside natural inhibitor, was identified as inhibitory against MurX/MraY (MurX in *M. tuberculosis*, MraY in other bacteria), which anchors completed UDP-MurNAc-pentapeptide to lipid carrier, forming lipid I (*Hotoda et al., 2003b*; *Yamaguchi et al., 1986*). Since then, extensive SAR studies have been conducted to enhance the potency of capuramycin analogues. Among thousands of compounds tested, SQ-641 (Fig. 9, compound 27) exhibited the most promising anti-mycobacterial activity with MIC <0.063 µg/mL despite the high $IC_{50}$ observed (550 ng/mL) (*Hotoda et al., 2003a*; *Hotoda et al., 2003b*). On the other hand, the parental compound, capuramycin, displayed significantly low $IC_{50}$ ($IC_{50}$ = 10 ng/mL) but higher MIC (MIC = 8 µg/mL). It is hypothesised that the increased anti-mycobacterial activity of SQ-641 could be attributed to higher lipophilicity, which also leads to low solubility and significantly impairs intracellular and *in vivo* activity (*Bogatcheva et al., 2011*). Several formulations were developed and examined to overcome this deficiency including $\alpha$-tocopheryl polyethylene glycol 1,000 succinate formulation, which was tested on the murine model and demonstrated better *in vivo* activity (*Nikonenko Boris et al., 2009*). However, the low drug loading of the formulation makes it impractical for human use. Other formulations such as phospholipid-based nanoemulsion and nanocarrier drug cocktail also enhance the *in vivo* activity of SQ-641 and warrant further search (*D'Addio et al., 2015*; *Nikonenko et al., 2014*).

Since the discovery of capuramycin, researchers have been focusing on synthesising analogues and conducting SAR without understanding the enzyme structure. The structure of mycobacterial MurX has not yet been solved, but the crystal structure of the homologue, MraY from *Aquifex aeolicus* (MraY$_{AA}$) has been resolved and exploited for rational inhibitor design based on protein-inhibitor interaction (*Ben et al., 2013*). The interaction between

**Figure 9** MurX inhibitors with the corresponding $IC_{50}$ value.

$MraY_{AA}$ and a collection of nucleoside natural product inhibitors was investigated. The studies reveal that nucleoside inhibitors induce similar conformational changes of MraY upon binding and expose cryptic binding sites not present in the apoenzyme (*Chung et al., 2016*). A uridine-binding pocket is formed in MraY to accommodate the common uridine moiety in all nucleosides upon binding. The uracil ring of uridine moiety is involved in an extensive hydrogen bonding network with loop C and loop D (*Chung et al., 2016*). The importance of uracil is also evident in SAR studies where major modification of uracil resulted in the loss of inhibition (*Heib et al., 2019*; *Wiegmann, Koppermann & Ducho, 2018*). Besides the common uridine-binding pocket, the uridine-adjacent pocket which varies in different inhibitors plays a significant role in inhibitor affinity and selectivity. For capuramycin, the uridine-adjacent pocket interacts with a hydroxyl group in 3,4-dihydroxy-3,4-dihyro-2H-pyran moiety by hydrogen bonding, which leads to a 10-fold difference in inhibition compared to analogues without the hydroxyl group (*Mashalidis et al., 2019*; *Muramatsu, Ishii & Inukai, 2003*). Furthermore, the unique caprolactam moiety of capuramycin occupies another hydrophobic binding pocket inaccessible to other nucleosides, contributing to its activity (*Mashalidis et al., 2019*). Interestingly, capuramycin does not form hydrogen bonding with loop E as other nucleoside inhibitors, which provides the opportunity for further optimisation (*Mashalidis et al., 2019*).

Another natural product, sansanmycin, was found to be active against *M. tuberculosis* with MIC = 9.5 μM (*Li et al., 2011*). Inspired by the favourable antimycobacterial effects, analogue library synthesis and screening were conducted to further enhance the anti-TB activity. The study successfully generated potent compounds (Fig. 9, compound 28) that demonstrated antimycobacterial activity at the nanomolar concentration (MIC = 37 nM). A MurX-MurG coupled assay displayed the addition of sansanmycin halting the formation of lipid II ($IC_{50}$ = 41 nM), suggesting the likely mechanism of action for sansanmycin (*Tran et al., 2017*). Another SAR study utilised the crystal structure of $MraY_{AA}$ to predict the mycobacterial MurX structure model and aim to enhance the potency of sansanmycin with the structure-based analogue design. The 18 compounds generated exhibited significant *in vitro* anti-mycobacterial activity (MIC = 5.9–192.3 nM) as well as activity against intracellular *M. tuberculosis* (MIC = 0.5–12.5 μM) (*Tran et al., 2021*). It is

also worth noting that the compound with the highest *in vitro* potency only has modest intracellular activity, possibly attributed to its lower membrane permeability. Subsequently, the analogues are screened for inhibitory activity against *M. tuberculosis* MurX. Intriguingly, it is discovered that compounds bearing amide linkage are significantly more potent than those with ester linkage despite similar *in vitro* and intracellular anti-mycobacterial activity ($IC_{50}$ = 20–60 nM *vs* $IC_{50}$ = 614–2,617 nM) (*Tran et al., 2021*). It is suggested that the ester-bearing compounds act as prodrugs and the ester moiety is hydrolysed during *in vitro* assay, while this hypothesis still requires further examination.

## Inhibitors of MurG & MurJ

In contrast to other members in the Mur enzyme family, of which extensive research has been conducted, there is limited information on MurG and MurJ, which are responsible for the membrane-associated stage of peptidoglycan synthesis. MurG, as a glycosyltransferase, which attaches the GlcNAc moiety of UDP-GlcNAc to MurNAc in lipid I, yielding lipid-linked $\beta$-(1,4) disaccharide peptidoglycan precursor, known as lipid II (*Mohammadi et al., 2007*). The cytoplasmic facing lipid II is then flipped to the periplasm by lipid II flippase, MurJ, and undergoes polymerisation and cross-linking (*Kuk, Mashalidis & Lee, 2017*). Despite several HTS studies for MurG inhibitors being completed in the early 2000s, they mainly focused on *E. coli* MurG, with the mode of action for the identified inhibitors remaining unknown (*Helm et al., 2003*; *Ramachandran et al., 2006*). A recent study utilised a computational drug discovery pipeline including homology modelling, pharmacophore generation, and virtual compounds screening to investigate potential *M. tuberculosis* MurG inhibitors. This pipeline successfully identified 10 potential lead compounds for *M. tuberculosis* MurG while further *in vitro* validation is not provided (*Saxena et al., 2018*). For MurJ, the current literature notably lacks published examples to date. Despite there being antibiotics that interact with lipid II, such as vancomycin and ramoplanin, it is demonstrated that these two antibiotics do not bind directly to MurJ (*Bolla et al., 2018*). In summary, despite the importance of MurG and MurJ in peptidoglycan synthesis, the inhibition of MurG and MurJ remain largely unexplored and therefore holds significant potential for future research in the development of antibacterial agents and combatting existing issues of drug resistance.

## Polymerisation and crosslinking

After the translocation of lipid II, polymerisation and crosslinking occur to form the rigid peptidoglycan structure. This process is facilitated by bifunctional enzymes PonA1 and PonA2 which compose the transglycosylase and transpeptidase activities. In fact, it is well established that this enzymatic process can be an ideal drug target as ß-lactams such as penicillin restricts this process to achieve antibacterial activity. Unfortunately, ß-lactam antibiotics have limited clinical significance in the current *M. tuberculosis* treatment regime mainly credited to the innate ß-lactamase (BlaC) encoded in mycobacteria (*Flores et al., 2005*). It has been demonstrated that ß-lactams co-delivered with ß-lactamase inhibitors increases activity compared to ß-lactam alone (*Li et al., 2018*). Furthermore, a high level of the atypical transpeptidation (3 → 3 linkage) which occurs in mycobacteria facilitated by

Compound 29
MIC = 0.1 µg/mL

**Figure 10 The optimised carbapenem with MIC value evaluated for *M. tuberculosis*.**

unique L,D-transpeptidases (Ldts), also contributes to resistance against ß-lactams through (*Gupta et al., 2010*).

Recently, carbapenem was discovered and became increasingly important in combatting drug-resistant bacteria. As a board spectrum antibiotic, carbapenem demonstrates the ability to inhibit *M. tuberculosis* PBPs (PonA1/PonA2). Meropenem, an FDA-approved carbapenem, combined with clavulanate, a BlaC inhibitor, achieves a low MIC in MDR-TB and non-replicating TB (*Hugonnet et al., 2009*). Intriguingly, despite possessing the ß-lactam ring, evidence shows that carbapenem is a poor substrate for BlaC with a slow hydrolysis rate which can minimize the impact made by BlaC (*Sun, Fan & Zhong, 2021*). Moreover, beyond the capability of inhibiting PBPs and BlaC, carbapenems also exhibit potency against Ldts which facilitates the unique 3 → 3 crosslinking during the non-replicative stage (*Cordillot et al., 2013*; *Dubée et al., 2012*). Thus, carbapenem impedes mycobacterial peptidoglycan synthesis at multiple steps, making it an ideal scaffold for further optimisation. In fact, imipenem is now included in MDR-TB and NTM treatment regimes improved clinical outcomes observed (*Chavan et al., 2020*).

A carbapenem optimisation study conducted in 2021 demonstrated that methyl substitution on C5 $\alpha$ greatly improves the potency against *M. tuberculosis* (MIC = 0.1 µg/mL) even in the absence of ß-lactamase inhibitor (Fig. 10, compound 29). The optimised compound fully acylates $Ldt_{MT2}$, proving the mode of action (*Gupta et al., 2021*). This result elucidates the potential of carbapenems as antimycobacterial agents and warrants further investigation.

Amongst all ß-lactams, carbapenems possess the most potent activity for inhibiting Ldts (*De Munnik et al., 2020*). Compounds active against $Ldt_{MT2}$ and BlaC were identified as hit compounds with $pIC_{50}$ ranging 5.55–7.99. Several new classes of Ldt inhibitors including $\alpha$-chloro ketones, maleimides, acrylamides, fumaryl amides, an ebsulfur analogue, isatins, and nitriles have now been reported. The most active compounds are those that covalently interacting with Cys354 of Ldt, which resides in the active site and is catalytically essential (*De Munnik et al., 2023*). To summarise, despite *M. tuberculosis*

possessing potent ß-lactamase and unique 3 → 3 linkage, the recent development of carbapenem and identification of other novel analogues exhibits promising developments towards pan-crosslinking inhibition of peptidoglycan, providing hope in the fight against drug-resistant TB.

## CONCLUSIONS

Peptidoglycan is an essential component of bacterial cell walls, and hence bacterial viability, with enzymes dedicated to its biosynthesis being therefore critical. The exclusiveness and essentiality of PG biosynthetic enzymes make them a very attractive targets for antibiotics development. Furthermore, the structural and functional relationships amongst several of these enzymes may offer the opportunity for simultaneous inhibition, leading to improved bactericidal potency and potentially circumventing the rapid development of drug resistance (*Silver, 2007*; *Silver & Bostian, 1993*). These properties of essential targets identify the pathway as an ideal candidate towards the development of effective and safe anti-Tb drugs.

An unresolved question relates to the observations that many potent inhibitors of enzyme function do not possess antibacterial effects. These compounds are hampered by their capacity for cell wall penetration, and/or efficient efflux systems in many bacterial pathogens, including *M. tuberculosis* (*Dinesh, Sharma & Balganesh, 2013*; *Hameed et al., 2014*; *Humnabadkar et al., 2014*; *Silver, 2006*). The bacteriostatic or bactericidal effect of these inhibitors is significantly improved in some bacteria when pre-exposed to cell envelop permeabilizers, demonstrating cell entry bottleneck (*Baum et al., 2007*; *Hameed et al., 2014*). Future endeavours could focus on improved drug delivery strategies. Cell entry can be achieved by chemical optimisation of the molecule promoting passive diffusion or conjugation with a moiety that promotes active uptake. A ''pro-drug'' strategy is widely used to with pharmacologically active molecules to mask hydrophilic groups and increase the permeability of the compounds. The pharmacological inactive groups are then removed by enzymatic or chemical transformation in the bacterial cytoplasm (*Rautio et al., 2008*).

*M. tuberculosis* represents a great many challenges for the development of novel chemotherapies directed to its elimination. The enormous volume of research conducted to date on the peptidoglycan synthesis pathway, and its inhibition, when taken together highlights opportunities for continued efforts and novel strategies for the development of effective inhibitors. Concepts of multi-target inhibitors, pan-transpeptidase inhibitors, and compound pro-drugging for improved permeability are future tasks which are reasonable, and highly desirable, goals for researchers in the field.

## ACKNOWLEDGEMENTS

We thank Dr. Emily Strong for reviewing the manuscript.

### Funding

Cheng-Yu Chiang is a recipient of a University of Queensland Graduate School Scholarship for Higher Degree Research. The funders had no role in study design, data collection and analysis, decision to publish, or preparation of the manuscript.

### Grant Disclosures

The following grant information was disclosed by the authors:
University of Queensland Graduate School Scholarship for Higher Degree Research.

### Competing Interests

The authors declare there are no competing interests.

### Author Contributions

- Cheng-Yu Chiang conceived and designed the experiments, performed the experiments, analyzed the data, prepared figures and/or tables, authored or reviewed drafts of the article, and approved the final draft.
- Nicholas P. West conceived and designed the experiments, authored or reviewed drafts of the article, and approved the final draft.

### Data Availability

   This is a literature review.

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
