# Peer review of "The fall of the mycobacterial cell wall: interrogating peptidoglycan synthesis for novel anti-TB agents"

_PeerJ, doi:10.7717/peerj.18404_

## Round 0.1 · original submission · Minor Revisions

Respected Authors,

I hope this message finds you well.

I am writing to you regarding your recently submitted critical work. We have received reports from our esteemed peers suggesting minor revisions to your work. In this regard, I kindly request you to address the reviewers' comments and provide point-by-point responses.

We eagerly anticipate receiving your revised work. Please do not hesitate to reach out to us if you have any questions or concerns.

Thank you for your attention to this matter.

Best wishes,
Dr. Nagendran Tharmalingam
Handling Editor

·

Basic reporting

The article "The fall of the mycobacterial cell wall: Interrogating peptidoglycan synthesis for novel anti-TB agents." is a nice review about the inhibitors of a selection of mycobacterial enzymes involved in the biosynthesis of peptidoglycan.
The English is fine and, in general, the article is well written and clear. The background is enough for the understanding of the whole article and references are appropriate.
I just give an advice to authors: mDAP is reported in the text different times (line 88, 136, 137 and 150), but never explained as diaminopimelic acid. I think that all the acronyms should be explained the first time that occur in the text, so that even less expert readers may immediately identify the meaning. if you can use the full name before acronyms it would be great.

Experimental design

The article is a review with a huge litterature research work. Not all the protein of Mur family of M. tuberculosis are reported in the text, but the one described are enough to give a full picture of the state of art of research up to now. It's a pity that the inhibitors of some Mtb Mur enzymes have not been identified yet, but the text compensate this lack with the analysis of some inhibitors of the hortologous enzymes of other organisms.

Validity of the findings

The review is based on papers mostly published in the last 15 year (2010-2023). Anyway, despite many articles are not so recent I found this article very interesting as recap of the research about Mtb Mur enzymes. It is also usefull to have all the informations in the same paper instead of searching many different articles on the web.

Additional comments

Just some small comments to improve the text:
- there is a misspelling in line 142.
- I would make a bit more clear the title of figure 10, maybe something like "... MIC value evaluated against Mtb"? Just because "MIC value Mtb" is not so nice to read.
- I would underline if the inhibitors reported in the different figures are developped against the enzymes of Mtb or hortologous enzymes.
Anyway great job, it is a very interesting and nice review.

·

Basic reporting

This review addresses a broad and cross-disciplinary topic that aligns well with the journal’s scope. It provides a comprehensive overview relevant to drug discovery, making it of wide interest. This review stands out by presenting updated insights and address emerging trends.

Experimental design

The survey methodology is generally consistent with providing comprehensive and unbiased coverage. However, the gaps I felt are mentioned in the review to ensure all relevant aspects of the subject are included. Further, the sources are also adequately cited, with appropriate use of quotations and paraphrasing, thus maintaining academic rigor and credibility.
The review is also organized logically and facilitates a smooth flow of information for enhanced readability.

Validity of the findings

No Comments

Additional comments

The manuscript by Chiang CY and West NP presents an insightful and thorough exploration of peptidoglycan synthesis in bacteria, with a specific focus on Mycobacterium species. The authors successfully illustrate the potential of targeting peptidoglycan synthesis as a strategy to inhibit mycobacterial growth. The manuscript is meticulously written, well-organized, and offers a comprehensive overview of peptidoglycan biosynthesis, highlighting its significance in developing novel anti-TB agents.
However, I recommend that the authors address the following points to enhance the quality and clarity of the manuscript:
1. Inclusion of updated information on MDR and XDR TB:
The introduction section would benefit from the inclusion of updated information regarding Multidrug-Resistant (MDR) and Extensively Drug-Resistant (XDR) tuberculosis. Specifically, the authors should consider adding recent data on treatment options, costs, and challenges associated with managing MDR and XDR TB. This addition will provide a more current and comprehensive context for the reader.
2. Revisions to figure labels:
The labeling of Figures 1 and 2 should be revised to improve clarity. Clear and precise labels are essential for enhancing the understanding of the visual data presented. The current labels are not clearly visible due its small size and revising them will contribute to better reader comprehension.
3. Inclusion of a comprehensive table of peptidoglycan synthesis inhibitors:
To further increase the manuscript's readability and utility, I suggest the inclusion of a table that lists all inhibitors of peptidoglycan synthesis. The table should include details such as the target enzymes, mode of action, and minimum inhibitory concentrations (MICs). This addition will serve as a quick reference for readers and will support the detailed discussions presented in the text.
4. Discussion on FDA-approved peptidoglycan synthesis inhibitors:
The manuscript would benefit from a brief discussion on FDA-approved inhibitors of peptidoglycan synthesis that are currently used in clinical treatment. Including this information will provide readers with an understanding of the practical applications of these inhibitors and their relevance in the clinical management of TB.

---

## Round 0.2 · accepted · Accept

Dear Dr. West,

Thank you for your submission to PeerJ.

I am writing to inform you that your manuscript - The fall of the mycobacterial cell wall: Interrogating peptidoglycan synthesis for novel anti-TB agents - has been Accepted for publication.

This is an editorial acceptance; publication depends on authors meeting all journal policies and guidelines.

Best wishes,
Dr. Nagendran Tharmalingam
Handling Editor.